# Effect of Sevoflurane on the Proliferation of A549 Lung Cancer Cells

**DOI:** 10.3390/medicina59030613

**Published:** 2023-03-20

**Authors:** Sangwon Yun, Kyongsik Kim, Keuna Shin, Hanmi Park, Sunyeul Lee, Yongsup Shin, Aung Soe Paing, Songyi Choi, Chaeseong Lim

**Affiliations:** 1Department of Anesthesiology and Pain Medicine, Chungnam National University College of Medicine, Daejeon 35015, Republic of Korea; 2Department of Anesthesiology and Pain Medicine, Chungnam National University Sejong Hospital, Sejong 30099, Republic of Korea; 3Research Institute for Medical Sciences, Chungnam National University, Daejeon 35015, Republic of Korea; 4Department of Anesthesiology and Pain Medicine, Chungnam National University Hospital, Daejeon 35015, Republic of Korea; 5Department of Surgery, 1000 Bedded Naypyitaw General Hospital, Naypyitaw 15011, Myanmar; 6Department of Pathology, Chungnam National University, Daejeon 35015, Republic of Korea

**Keywords:** sevoflurane, lung cancer, A549 cells

## Abstract

*Background and Objectives*: Sevoflurane has opposing effects on cancer progression, depending on its concentration and the cancer type. This study investigated the effects of sevoflurane on the proliferation of A549 lung cancer cells. *Materials and Methods*: In vitro, the number of A549 cells exposed to different concentrations of sevoflurane was counted. The size and weight of tumors from a xenograft mouse model exposed to air or sevoflurane were measured in vivo experiments. Additionally, hematoxylin and eosin staining and immunohistochemical detection of Ki-67 in the harvested tumor tissues were performed. *Results*: A total of 72 culture dishes were used and 24 dishes were assigned to each group: Air group; 2% Sevo group (air + 2% sevoflurane); and 4% Sevo group (air + 4% sevoflurane). The number of A549 cells in the 2% Sevo group was less than that in the Air and 4% Sevo groups (Air: 7.9 ± 0.5; 0.5, 2% Sevo: 6.8 ± 0.4, 4% Sevo: 8.1 ± 0.3; *p* = 0.000). The tumor size was not significantly different between the two groups (Air: 1.5 ± 0.7, 2% Sevo: 2.4 ± 1.9; *p* = 0.380). *Conclusions*: The in vitro data showed that sevoflurane inhibited the proliferation of A549 lung cancer cells in a concentration-specific manner. However, the in vivo data showed no correlation between sevoflurane exposure and A549 cell proliferation. Thus, further research is required to understand fully the effects of sevoflurane on cancer progression and to reconcile differences between the in vitro and in vivo experimental results.

## 1. Introduction

Lung cancer is one of the five most common cancers and the leading cause of cancer-related death, with an incidence rate of 27.6 per 100,000 and a mortality rate of 15.7 per 100,000 persons in Korea in 2018 [1,2]. Although several modalities such as surgery, radiotherapy, and chemotherapy have been rapidly developed, with significant advances, surgical resection of a tumor remains the cornerstone for the treatment of early-stage lung cancer [3,4,5]. Because anesthesia is essential for surgery, anesthesiologists must consider the effects of anesthetics on cancer progression and carefully choose the appropriate anesthetic for cancer patients that will result in optimal short- and long-term outcomes.

Several studies have reported significant suppression of host immunity by the surgical stress response with subsequent neuroendocrine, metabolic, and inflammatory changes during the perioperative period, which may influence malignant cell development [6,7]. Anesthetic factors also have an important role in cancer growth and invasion by altering the immune system [8,9]. Nevertheless, there are several conflicting results showing that sevoflurane affects the regulation of anti-cancer relevant signaling pathways via microRNAs, matrix metalloproteinases, cell apoptosis, and transcription factors, resulting in antiproliferation, antimigration, and the anti-metastasis of cancer [10,11,12]. There is a lack of data on the effects of inhalation agents on the proliferation, migration, and metastasis of cancer cells [13].

Sevoflurane is a commonly used inhalational anesthetic agent in modern anesthesia practice. There have been concerns regarding the potential link between sevoflurane exposure and cancer development, particularly in animal studies. However, the evidence on the association between sevoflurane exposure and cancer in humans is limited and conflicting. Some studies suggest a possible link between sevoflurane exposure and increased risk of cancer recurrence and mortality in patients undergoing cancer surgery [6,7]. Other studies have not found any significant association [14]. It is important to note that cancer development is a complex process that involves many factors, including genetic predisposition, lifestyle, environmental exposure, and other medical conditions. While some studies [10,11,12] have suggested a possible link between sevoflurane exposure and cancer, it is difficult to establish a direct cause-and-effect relationship.

During lung cancer surgery, cancer cells are exposed to sevoflurane because sevoflurane is an inhalational anesthetic. The effect of sevoflurane on lung cancer cells will be more significant than on other cancers. This study investigated the effect of sevoflurane on the proliferation of A549 lung cancer cells in vitro and in vivo. To this end, the number of A549 cells exposed to sevoflurane was counted and the tumor size and weight from a xenograft mouse model were measured. Additionally, pathological assessment of tumor tissues with hematoxylin and eosin staining and immunohistochemical detection of Ki-67 were performed.

## 2. Materials and Methods

### 2.1. Cell Preparation and Culture

A549 adenocarcinomic human alveolar basal epithelial cells were obtained from the American Type Culture Collection (Manassas, VA, USA). They were cultured in Roswell Park Memorial Institute (RPMI) 1640 medium supplemented with 10% fetal bovine serum, 100 U/mL penicillin, and 100 μg/mL streptomycin at 37 °C in a 5% CO_2_ incubator.

### 2.2. In Vitro Study

#### 2.2.1. Sevoflurane Exposure with Oxygen (Figure 1)

On Day 1, 1 × 10^5^ A549 cells were seeded in 100 mm cell culture dishes (VWR International, Leicestershire, UK) at 37 °C in a 5% CO_2_ incubator. The A549 cells were exposed to different concentrations of sevoflurane (0%, 2%, 4%) with oxygen for 3 days. Cell culture dishes in a hand-made container with a gas inlet and outlet valves were placed in a 38 °C water bath. Depending on the condition, the different concentrations of sevoflurane and 1 L/min oxygen were delivered to the container. The appropriate concentration of sevoflurane was adjusted by turning the vaporizer dial based on the gas flow analyzer (Datex, Helsinki, Finland). The exposure experiment was conducted three times a week, for 1 h each time. Daily images were obtained by a microscope camera. The confluency of A549 cells reached about 75% on Day 4.

**Figure 1 medicina-59-00613-f001:**
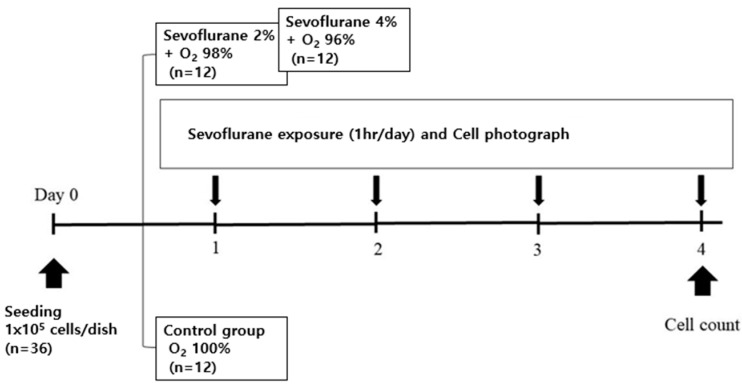
Schematic illustration for the in vitro study with oxygen.

#### 2.2.2. Sevoflurane Exposure with Air (Figure 2)

With the exception of replacing oxygen with air, the experimental protocol and environment were the same as in Section 2.2.1. Twice as many experiments were performed with air as with oxygen, so the total number is 72.

**Figure 2 medicina-59-00613-f002:**
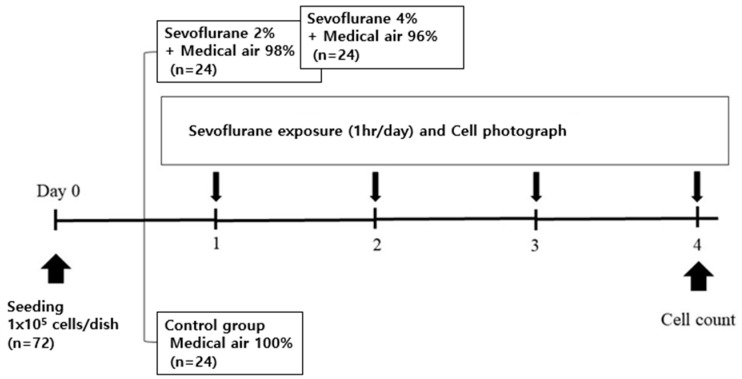
Schematic illustration for the in vitro study with air.

#### 2.2.3. Cell Viability Analysis

On Day 5, the extent of proliferation was evaluated by counting the number of A549 cells with a hemocytometer and analyzing the images obtained from a microscope photograph in a low power field (40×).

### 2.3. In Vivo Study

#### 2.3.1. Animal Preparation

BALB/c nude mice (male, 6 weeks old) were acclimated to new cages and environments for 1 week. The animal cages were maintained at approximately 40–60% humidity at 20–26 °C, under a 12 h light/dark cycle with frequent ventilation. Food and water were provided ad libitum. The animal experimental protocols were approved by the Institutional Animal Care and Use Committee of Chungnam National University Hospital on 27 December 2019 (No. CNUH-020-A0024; Daejeon, Republic of Korea).

#### 2.3.2. Xenograft Mouse Model

To establish a xenograft model, 200 μL phosphate-buffered saline (PBS, pH 7.0) containing 5 × 10^6^ A549 cells was prepared and subcutaneously injected into the right thigh of each mouse.

#### 2.3.3. Sevoflurane Exposure with Air (Figure 3)

Ten mice were randomly divided into two groups: Air group (exposure to only air, *n* = 5) and 2% Sevo group (exposure to 2% sevoflurane and air, *n* = 5). The xenograft mice in the container in the 38 °C water bath were exposed to 1 L/min of air or 2% sevoflurane with 1 L/min of air. The exposure experiment in the two groups was performed three times a week for 5 weeks, for 1 h each time.

**Figure 3 medicina-59-00613-f003:**
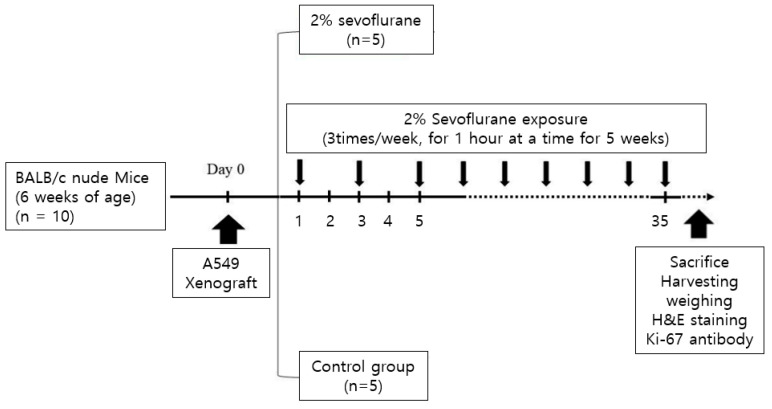
Schematic illustration for the in vivo study with air.

#### 2.3.4. Measurement of Tumor Size and Mouse Weight

During the experimental period, the tumor size and mouse weight were measured. The length and width of the tumor were measured by a caliper and the tumor volume was calculated as follows [14,15]:*tumor volume* (cm^3^) = *length* × *width*^2^ × 0.5

All mice were weighed three times a week.

#### 2.3.5. Sacrifice, Harvesting, and Weighing the Resected Mass

The tumor became too large for the xenograft mice to survive at 6 weeks from initiation of the animal study. After conducting the sevoflurane exposure experiment for 5 weeks, all mice were sacrificed by placing them in a sealed chamber filled with CO_2_ according to the American Veterinary Medical Association Guidelines. The tumor tissues were removed, the tumor weight was measured, then the tumor tissues were cut into 1 cm pieces and fixed in 4% paraformaldehyde for pathological analysis.

#### 2.3.6. Hematoxylin and Eosin Staining and Immunohistochemistry Cell Preparation

Sections (3 μm thick) of formalin-fixed paraffin-embedded blocks of tumors from each group were stained with hematoxylin and eosin (H&E). Because two sections were obtained from each mouse, 10 slides per group were obtained. Immunohistochemical detection of Ki-67 protein was also performed on 10 tumor specimens from each group with Ki-67 antibodies (1:200, MIB-1; DAKO, Glostrup, Denmark).

#### 2.3.7. Analysis of H&E Staining and Ki-67 Immunohistochemistry

Cell swelling, nuclear swelling, karyolysis, karyorrhexis, nuclear pyknosis, pale eosinophilic cytoplasm, and the presence and absence of cytoplasmic vacuoles may be visualized in necrosis stained with H&E [16]. Because necrotic cells are unable to maintain cell membrane integrity, their contents often leak out and elicit inflammation in the surrounding tissue. Therefore, the area of necrotic tissues that showed these characteristics and the area of entire tissue in each slide were measured using a digital microscope viewer (3D Histech, Budapest, Hungary) in a high-power field (100×). The degree of tissue necrosis was calculated by dividing the area of the necrosis tissues by the area of the entire stained tissues. The slides of tumor tissues stained with Ki-67 antibody were observed in a high-power field (400× magnification). Positive nuclear staining of tumor cells was of varying intensity, and any staining of nuclei was considered positive. The Ki-67-positive index was defined as the number of Ki-67-positive tumor cells divided by the sum of Ki-67-positive and Ki-67-negative tumor cells after manual counting of at least 500 tumor cells above per specimen.

### 2.4. Statistical Analyses

The normal or non-normal distribution of all data was evaluated using the Shapiro–Wilk test. When the normality was satisfied, statistical significance was evaluated using the *t*-test or analysis of variance according to the number of groups. The Mann–Whitney U test or Kruskal–Wallis test was used to determine statistical significance if the data were not normally distributed. All statistical analyses were performed using R software version 4.1.0 (R project for Statistical Computing, Vienna, Austria). Statistical significance was defined as *p* < 0.05.

## 3. Results

### 3.1. In Vitro Study

To assess the effect of sevoflurane on A549 lung cancer cells in vitro, two quantitative methods were used to count the cells: a hemocytometer and a microscope image. In the first experiment, 36 culture dishes seeded with A549 cells were divided equally into three groups and exposed to different conditions (O_2_ group: O_2_ only, 2% Sevo group: O_2_ + 2% sevoflurane, 4% Sevo group: O_2_ + 4% sevoflurane). The number of A549 cells cultured with oxygen was significantly increased with increasing the concentration of sevoflurane (O_2_: 6.3 ± 0.3, 2% Sevo: 7.0 ± 0.7, 4% Sevo: 7.3 ± 0.6; *p* = 0.000). When the cells were observed by microscope digital images, it was visually confirmed that the number of A549 cells increased as the concentration of sevoflurane increased (Figure 4a,c). However, the results of the second experiment, in which the oxygen was replaced with air, were unexpected. A total of 72 culture dishes were used and 24 dishes were assigned to each group: Air group (air only); 2% Sevo group (air + 2% sevoflurane); and 4% Sevo group (air + 4% sevoflurane). The number of A549 cells in the 2% Sevo group was less than that in the Air and 4% Sevo groups (Air: 7.9 ± 0.5, 2% Sevo: 6.8 ± 0.4, 4% Sevo: 8.1 ± 0.3; *p* = 0.000). In addition, the number of cells in the 2% Sevo group was less than that in the other two groups under gross visual observation using a microscope digital image (Figure 4b,c).

### 3.2. In Vivo Study

In the animal experiment, 10 xenograft mice were equally assigned to two groups (Air and 2% Sevo group) and there were no mortalities. The tumor in the xenograft mouse model was observed 2 weeks after the injection of A549 cells. The tumor size gradually increased over time. There were no statistically significant differences between the initial weight (Air: 25.8 ± 1.2, 2% Sevo: 25.6 ± 1.0; *p* = 0.763) and the final weight (Air: 25.9 ± 2.1, 2% Sevo: 26.2 ± 2.0; *p* = 0.868) of mice between the Air and 2% Sevo groups. The tumor size was also not significantly different between the two groups (Air: 1.5 ± 0.7, 2% Sevo: 2.4 ± 1.9; *p* = 0.380) (Table 1). Subsequently, the size of the harvested tumors from the xenograft mouse model in the Air group looked similar to each other with the naked eye; however, they differed from each other in size in the 2% Sevo group (Figure 5). The weight of the resected tumors was 4.6 ± 0.9 g in the Air group and 4.6 ± 3.3 g in the 2% Sevo group (*p* = 0.978), showing no statistically significant difference between the groups.

### 3.3. The Pathological Experiments with Resected Tumor

Each of the 10 slides (two slides were obtained from each mass) from the Air group and 2% Sevo group were used. The areas of necrotic tissue and the entire amount of tissue from each specimen were stained with H&E (Figure 6a). The necrosis ratio was 16.7 ± 3.2 in the Air group and 16.7 ± 2.9 in the 2% Sevo group (*p* = 0.972), with no statistically significant difference between groups (Table 2). Each of the 10 sections with good quality Ki- 67 staining from the two groups were assessed (Figure 6b). The Ki-67-positive index, defined as the extent of Ki-67-positive staining in the entire number of cells, was 86.8 (84.4; 88.8) in the Air group and 87.6 (80.3; 88.6) in the 2% Sevo group (*p* = 0.925). The index represented the degree of proliferation of A549 cells (Table 2).

## 4. Discussion

In the cell viability experiment with sevoflurane and oxygen, the number of A549 cells tended to increase significantly as the concentration of sevoflurane increased. One study reported that sevoflurane exposure inhibits the malignant potential of head and neck squamous cell carcinoma in proportion to the increase in sevoflurane concentration [17]. The results from the first in vitro experiment showed that sevoflurane may affect cancer proliferation at the cellular level and that the concentration of sevoflurane may be an important factor for cancer proliferation. Jeffrey [18] investigated the effect of oxygen toxicity on cell culture experiments, showing that a lethal dose of oxygen exposure to cultured cells causes cell death. In addition, increased reactive oxygen intermediates and the concentration of oxygen are closely related to cell death, resulting from chromosomal breakage and cell cycle arrest [18,19]. In a clinical situation such as cancer surgery, it is uncommon to use a high concentration of oxygen together with sevoflurane.

It is important to note that the effects of oxygen concentration on cancer cells are complex and multifaceted. In some cases, high oxygen concentrations may actually promote the growth of cancer cells, particularly in tumors that are adapted to low oxygen environments. There is some evidence to suggest that high oxygen concentrations may inhibit the growth of certain types of cancer cells in vitro. Therefore, instead of concluding that the results of the oxygen-based study (left figure in Figure 4c) were purely due to the effect of sevoflurane, an air-based experiment was added. Therefore, additional cell viability experiments are planned to exclude the effects of oxygen.

The second cell viability experiment showed that A549 cells in the 2% Sevo group proliferated more slowly than those in the Air and 4% Sevo groups, indicating that the toxicity of oxygen contributed primarily to the proliferation of A549 cells in the first cell viability experiment. Sevoflurane can exert tumorigenic effects on different human cancer cell lines. One study reported that sevoflurane promotes breast cancer cell proliferation, migration, and invasion, but not in a dose-dependent manner [20]. Another study showed that sevoflurane can promote the expansion of glioma stem cells via the upregulation of hypoxia-inducible factors (HIFs) [21]. However, whether the common volatile anesthetic, sevoflurane, suppresses A549 lung cancer cell proliferation by regulating apoptosis [22,23] and inhibiting HIF-1α [24] remains a subject of debate. It seems that the effect of sevoflurane on cancer proliferation varies depending on the cancer type and the concentration of sevoflurane. From these perspectives, the second experiment could lead to the conclusion that the direct exposure of sevoflurane has an inhibitory effect on A549 lung cancer cell proliferation in a dose-dependent manner. One study by Liang [24] concluded that exposure to 2.5% and 3.5% but not 1.5% sevoflurane led to suppression of hypoxia-induced proliferation and metastasis of lung cancer cells. Therefore, further research is essential to clarify the changes in molecular mechanisms depending on the concentration of sevoflurane.

The proliferation of A549 lung cancer cells represented by changes in tumor size and weight was not statistically correlated with the exposure of sevoflurane in the animal experiment. Several studies using a xenograft mouse model have demonstrated the suppressing effect of sevoflurane on proliferation in human ovarian cancer and glioma via various signaling pathways [25,26]. Conflicting data have shown that sevoflurane promotes the tumor growth of a xenograft mouse implanted with human glioblastoma cancer cells by increasing the expression of cell surface protein 44 [27]. Sevoflurane has also been shown to suppress A549 lung cancer cell proliferation by regulating the relevant anti-cancer signaling pathway, such as matrix metalloproteinase and HIF-1α [10]. Because the concentration of sevoflurane in the current animal experiment differed from that of previous studies, we concluded that 2% sevoflurane had no effect on A549 lung cancer cell proliferation in vivo. Even though well-designed in vitro experiments provide valuable data, they may not reflect the complexity and delicate and complicated microenvironments and homeostatic mechanisms of intact mammals [28]. This may explain why the xenograft mouse experiment did not produce the same results as the in vitro experiment in this study.

In the H&E staining experiment, the ratio of necrosis did not show a statistically significant difference in the presence or absence of sevoflurane exposure. Solid tumor necrosis commonly results from inadequate vascularization and metabolic stress caused by aggressive tumor development [29]. Because A549 lung cancer cells grow in a spherical shape, the ratio of necrosis to the entire tissue could be thought of as the degree of tumor proliferation. In addition, the nuclear protein Ki-67, which is widely used as a direct proliferation marker of human tumor cells [30,31], was used for immunohistochemistry of the resected tumor tissues. Because Ki-67 protein is expressed during cell cycle phases G1, S, G2, and M but not found during G0, a Ki-67-positive index can represent the proportion of currently proliferating cancer cells [32]. Even though the Ki-67-positive index of the Air group and 2% Sevo group was more than 80%, sevoflurane exposure did not influence the active proliferation of A549 cells. Because few information studies about the relationship between sevoflurane and Ki-67 expression are available, future studies are needed to investigate the molecular mechanisms underlying the effect of sevoflurane on the proliferation of A549 lung cancer cells.

This study had several limitations. First, if an additional cell viability assay, such as the adenosine triphosphate assay or 3-(4,5-dimethylthiazol-2-yl)-2,5-diphenyltetrazolium bromide (MTT) assay had been applied, more reliable and complementary results might have been produced [33]. This is because sevoflurane inhibits the proliferation of A549 cells in a concentration-dependent manner [34]. Second, cancer invasion into muscles and bones has been found in some xenograft mice. Since the measurement of tumor size using a caliper is error-prone [35], weighing extracted tumors was performed to evaluate the degree of cancer proliferation. However, the remaining muscles and connective tissues attached to the tumor could have caused an error in tumor weight. Third, the evaluation of Ki-67 immunohistochemistry was performed by only one researcher. Manual counting on camera-captured images of tumor hot spots is the most accurate, reproducible, and practical method compared to automated counting and eyeballing [36]. However, if high interobserver agreement is guaranteed, more reliable evidence for the effect of sevoflurane on cancer proliferation is possible.

The use of sevoflurane in anesthesia should be balanced against the potential risks and benefits for each individual patient. It is important to discuss any concerns about sevoflurane and cancer risk with a qualified healthcare provider. One study published in 2013 found that exposure to sevoflurane may promote the growth and proliferation of certain types of cancer cells, including lung cancer cells [8]. The researchers conducted in vitro experiments using human lung adenocarcinoma cell lines and found that exposure to sevoflurane led to increased cell proliferation and resistance to chemotherapy drugs. Another study published in 2018 investigated the effect of sevoflurane exposure on lung cancer cells [22]. The researchers used a quantitative real time polymerase chain reaction (qRT-PCR) analysis and found that exposure to 3% sevoflurane suppressed miRNA interference in cancer cells.

However, it is important to note that these studies were conducted in vitro or in animal models, and the clinical relevance of these findings in humans is not yet fully understood. Further research is needed to better understand the potential implications of sevoflurane exposure on lung cancer development and progression. A clinical study published in 2022, which retrospectively analyzed more than 1500 lung cancer surgeries, also proved that there was no difference in the long-term prognosis of lung cancer between propofol-based TIVA and sevoflurane-based inhalation anesthesia [37].

## 5. Conclusions

The results from the in vitro experiments showed that sevoflurane had dose specific inhibitory effects on A549 lung cancer cell proliferation, especially under exposure of 2% sevoflurane. However, the xenograft and pathologic experiments showed that there was no significant effect of sevoflurane on A549 lung cancer cell proliferation in vivo. Further research is needed on why only 2% sevoflurane showed an effect that was not found in 4% in vitro. Given that no effect of sevoflurane was shown in the in vivo experiment, there is no need to feel burdened with the use of sevoflurane in lung cancer surgery in clinical practice.

## Figures and Tables

**Figure 4 medicina-59-00613-f004:**
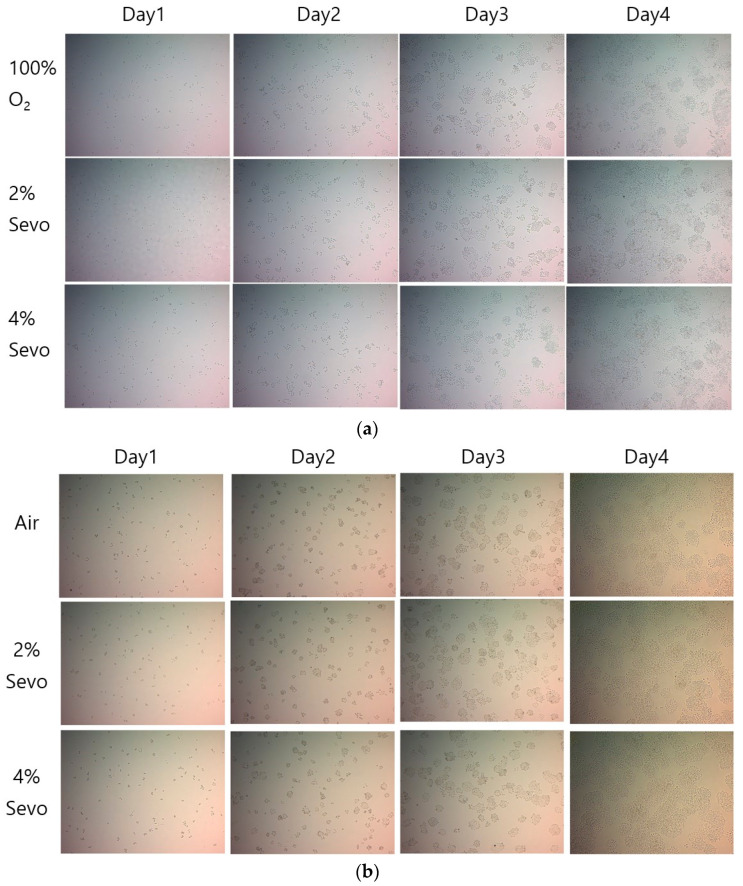
A549 cancer cell proliferation after sevoflurane exposure in vitro studies. (**a**) Representative microscopic images when A549 cells were exposed to 0%, 2%, or 4% sevoflurane with oxygen or (**b**) with air for three days (×40). (**c**) A comparative analysis of the number of A549 cells exposed to the different concentration of sevoflurane with oxygen (left figure, total *n* = 36) or air (right figure, total *n* = 72) on day 4. Data are presented as the mean ± SD. * represents *p* < 0.01, compared with 0% sevoflurane.

**Figure 5 medicina-59-00613-f005:**
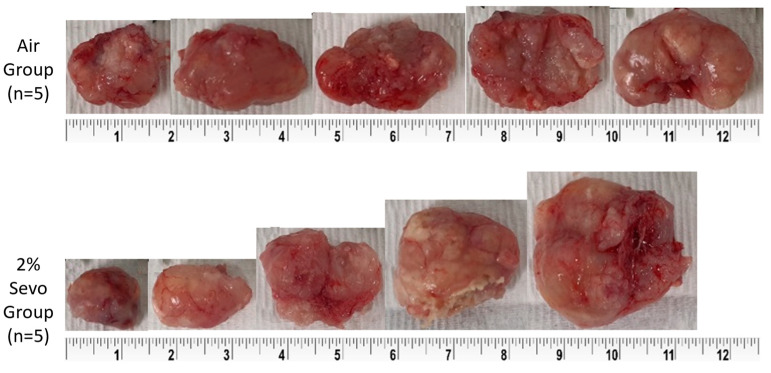
Comparison of the harvested tumor size in Air group and 2% Sevo group.

**Figure 6 medicina-59-00613-f006:**
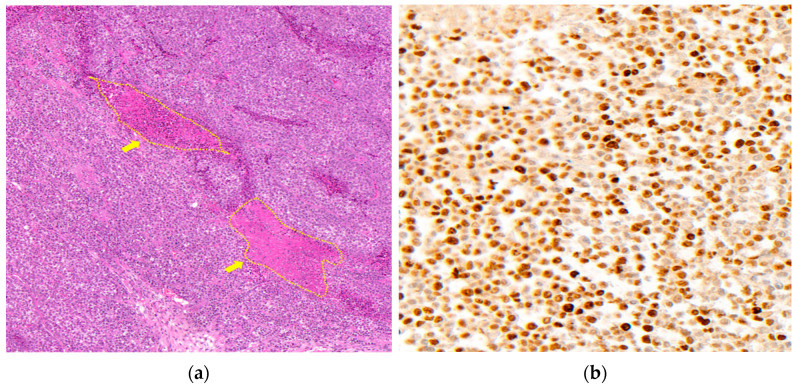
Pathological analysis of resected tumor tissues. (**a**) Representative example of H&E staining of A549 tumor tissues with necrosis (×100). Yellow arrows indicate necrosis area with inflammation. (**b**) Representative example of Ki-67 immunohistochemistry (×400). Tumor cells with brown colored nucleus were counted as positive.

**Table 1 medicina-59-00613-t001:** Comparison of the measured values from the xenograft mouse model.

	Air Group (*n* = 5)	2% Sevo Group (*n* = 5)	*p* Value
Initial weight (g)	25.8 ± 1.2	25.6 ± 1.0	0.763
Final weight (g)	25.9 ± 2.1	26.2 ± 2.0	0.868
Weight change (g)	−0.4 ± 1.4	0.2 ± 1.3	0.528
Tumor size (cm^3^)	1.5 ± 0.7	2.4 ± 1.9	0.38
Tumor weight (g)	4.6 ± 0.9	4.6 ± 3.3	0.978

Data are presented as the mean ± SD.

**Table 2 medicina-59-00613-t002:** Comparison of necrosis ratio and Ki-67 positive index of tumor tissues between Air group and 2% Sevo group.

	Air Group (*n* = 10)	2% Sevo Group (*n* = 10)	*p* Value
Necrosis ratio (%)	16.7 ± 3.2	16.7 ± 2.9	0.972
Ki-67 positive index (%)	86.8 (84.4; 88.8)	87.6 (80.3; 88.6)	0.925

Data are presented as the mean ± SD or the median.

## Data Availability

The data presented in this study are available on request from the corresponding author.

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
