# Peer review of "Effect of Sevoflurane on the Proliferation of A549 Lung Cancer Cells"

_medicina, 2023, doi:10.3390/medicina59030613_

Round 1

Reviewer 1 Report

Dear Authors,

I read the manuscript entitled "Effect of sevoflurane on the proliferation of A549 lung cancer cells" with great interest.

Even though this is a relatively small trial with results which must be interpreted with great caution I feel it is a nice "proof of concept trial".

That fact that in vivo results do not reflect in vitro scenario is of utter importance to the clinical oncology practice.

I do not have any major concerns in regard to the trial, and I do feel it should be published.

Reviewer.

Author Response

Thank you. Please take a look at the answer file and the renewed manuscript.

Reviewer 2 Report

The introduction needs enhancement. Grammar errors must be checked and a comprehensive edition is needed.

Differences between cell lines and tumor experiments results must be thoroughly discussed and deeper insights into mechanisms advised. 

Author Response

(The authors gave the same response as above.)

Reviewer 3 Report

1. what is the technically differently between air and oxygen, which factor resulting that. why 2%group in air cause a significant result. it is lacking of a reasonably explain

2.you need to find out other pathway expect insignificance Ki67

3.most important thing is that what is the clinical meaning of sevoflurane effecting lung cancer proliferation, effecting the anesthesia drug select in lung cancer surgery ?

  •  

Author Response

(The authors gave the same response as above.)

Round 2

Reviewer 2 Report

Thanks for making the suggested changes.